# Oxidized-Multiwalled Carbon Nanotubes as Non-Toxic Nanocarriers for Hydroxytyrosol Delivery in Cells

**DOI:** 10.3390/nano13040714

**Published:** 2023-02-13

**Authors:** Panagiota Zygouri, Antrea M. Athinodorou, Konstantinos Spyrou, Yannis V. Simos, Mohammed Subrati, Georgios Asimakopoulos, Konstantinos C. Vasilopoulos, Patra Vezyraki, Dimitrios Peschos, Konstantinos Tsamis, Dimitrios P. Gournis

**Affiliations:** 1Department of Materials Science and Engineering, University of Ioannina, 45110 Ioannina, Greece; 2Nanomedicine and Nanobiotechnology Research Group, University of Ioannina, 45110 Ioannina, Greece; 3Department of Physiology, Faculty of Medicine, School of Health Sciences, University of Ioannina, 45110 Ioannina, Greece

**Keywords:** multi-walled carbon nanotubes, oxidation, hydroxytyrosol, cytotoxicity, antioxidant

## Abstract

Carbon nanotubes (CNTs) possess excellent physicochemical and structural properties alongside their nano dimensions, constituting a medical platform for the delivery of different therapeutic molecules and drug systems. Hydroxytyrosol (HT) is a molecule with potent antioxidant properties that, however, is rapidly metabolized in the organism. HT immobilized on functionalized CNTs could improve its oral absorption and protect it against rapid degradation and elimination. This study investigated the effects of cellular oxidized multiwall carbon nanotubes (oxMWCNTs) as biocompatible carriers of HT. The oxidation of MWCNTs via H_2_SO_4_ and HNO_3_ has a double effect since it leads to increased hydrophilicity, while the introduced oxygen functionalities can contribute to the delivery of the drug. The in vitro effects of HT, oxMWCNTS, and oxMWCNTS functionalized with HT (oxMWCNTS_HT) were studied against two different cell lines (NIH/3T3 and Tg/Tg). We evaluated the toxicity (MTT and clonogenic assay), cell cycle arrest, and reactive oxygen species (ROS) formation. Both cell lines coped with oxMWCNTs even at high doses. oxMWCNTS_HT acted as pro-oxidants in Tg/Tg cells and as antioxidants in NIH/3T3 cells. These findings suggest that oxMWCNTs could evolve into a promising nanocarrier suitable for targeted drug delivery in the future.

## 1. Introduction

Since the discovery of carbon nanotubes (CNTs) by Iijima and co-workers in 1991 [1], a vast scientific community from diverse research areas has undertaken efforts to explore this peculiar one-dimensional nanomaterial that possesses extraordinary properties. Their shape consists of graphitic walls ranging from single graphene (SWCNTs) to multiple graphene sheets (MWCNTs) [2], wrapped in cylinders of a few nanometers in diameter and several micrometers in length. This unique shape, alongside the p-p conjugated system and the small nanometer size of the CNTs, generates a series of impressive structural, mechanical, optical, and electronic properties [3]. Among them are excellent electrical and thermal conductivity and exceptional tensile strength combined with high elasticity, chemical stability, and an ultrahigh surface area, while the surface can be easily tailored and functionalized with a wide variety of biomolecules for specific biomedical targets [4]. As a consequence, CNTs have found a dominant place in many applications [5], such as in transistors in microelectronics [6,7], coatings and fillers in nanocomposites [8], energy storage devices [9], biosensors [10], and medical devices [11].

Recently, there has been a revolution in how carbon nanotubes are utilized as effective vehicles for drug delivery due to the fact that they have the ability to adsorb or conjugate a wide variety of pharmaceutical drugs, biomolecules, proteins, enzymes, or DNA. Drug nanocarriers are categorized as organic (e.g., liposomes and solid lipid nanoparticles) and inorganic (e.g., gold, silica, carbon nanoparticles). Organic nanocarriers are, in general, non-toxic and have the potential to encapsulate and deliver a wide range of pharmaceutical drugs [12]. On the other hand, inorganic nanocarriers have raised issues regarding their potential toxicity [13]. Amongst inorganic nanocarriers, CNTs have attracted significant attention for drug delivery [14]. CNTs have been examined as potential carriers for anticancer drugs, DNA/RNA, enzymes, antibodies, and antibiotics [15,16,17]. The greatest obstacle to the usefulness of such molecules and subsequently their targeted bio-applications is that CNTs are highly hydrophobic, which is a limiting factor for drug delivery systems. In order for CNTs to be used as a vehicle for drugs and reach cells, they should be first functionalized for two main reasons: (a) to create the appropriate functionalities for the drugs to absorb and (b) to increase the solubility in polar/aqueous media. An efficient way to achieve this is via the oxidation of MWCNTs by various reagents [18].

Nanotechnology allows scientists to modify the physical, chemical, and biological characteristics of particles at the nanoscale and to develop nano-carriers for drug delivery and diagnostic applications. Nano-enhanced drugs are designed to maximize the delivery of non-water-soluble drugs, act directly on specific cells or tissues, and overcome problems such as drug transport at the brain–blood barrier (BBB) [19]. Hydroxytyrosol (HT) is a phenolic compound found in olive oil, olive leaves, and wine and exerts a broad range of potential pharmacological activities, such as antioxidant, anti-inflammatory, neuroprotective, antimicrobial, cardioprotective, and anticancer effects [20,21,22,23]. Due to its medicinal benefits, various methodologies have been proposed for the extraction of HT from olive oil or olive leaves and production by chemical synthesis or by microorganisms through the application of biotechnological approaches [24,25].

The absorption of HT is a dose-dependent procedure that takes place in the intestine and undergoes intestinal and hepatic metabolism [26]. The absorption efficiency of HT ranges between 75 and 100%. HT and its metabolites are distributed extensively to the body tissues and can even cross the BBB. This property could explain the advantageous health effects of HT [27,28]. However, HT absorption is a rapid procedure leading to maximum plasma levels a few minutes after intake, while the complete elimination of HT and its metabolites from the body occurs after 6 h in humans [29]. Due to its potent and rapid metabolism, HT exhibits a plasma half-life of 1–2 min [28].

Bioavailability is a prerequisite for any compound to exhibit a biological effect. Thus, a biocompatible nanocarrier that protects and increases the drug’s half-life could act multiplicatively and enhance its biological activity. The toxic effects of CNTs have been studied for the past several years, and it was concluded that CNTs’ toxicity is related to the types of nanotubes, the type of functionalization, and the purity of the final product [14]. HT is a molecule that is rapidly metabolized in the body, so it is possible that effective therapeutic blood and tissue concentrations are not achieved. This study aimed to utilize newly synthesized oxMWCNTs as nanocarriers for HT, assess their physicochemical characteristics, and determine how they interact with cells.

## 2. Materials and Methods

### 2.1. Materials

The pristine multi-walled carbon nanotubes, MWCNTs (carbon purity > 95%), were purchased from Sigma-Aldrich. For the oxidation process, two chemicals were used, 65% nitric acid (HNO_3_) and 95–97% sulfuric acid (H_2_SO_4_), which were supplied by Merck.

### 2.2. Oxidation of Multi-Walled Carbon Nanotubes-oxMWCNTs

First, 100 mg of pristine MWCNTs was added to a mixture of 30 mL H_2_SO_4_ (95–97%) and 10 mL HNO_3_ (65%) (3:1 *v*/*v*%). The dispersion flask was placed in a sonicator bath for 3 h. The resulting oxidized MWCNTs (oxMWCNTs) were separated from the mixture by centrifugation, which was followed by four washings with distilled water. The obtained oxidized material was dried in ambient conditions on a glass plate (Figure 1).

### 2.3. Decoration of oxMWCNTs with Hydroxytyrosol-oxMWCNTs_HT

Τhe ability of the oxidized carbon nanotubes to be an effective carrier in drug delivery systems is examined in this study. For this purpose, 50 mg of HT and 50 mg of oxMWCNTs were added, each one in 50 mL of distilled water. The aqueous dispersions were mixed and stirred for 24 h. The precipitate was collected after centrifugation and several washings with distilled water. The final product was air-dried on a glass plate (Figure 2).

### 2.4. Characterization Techniques for oxMWCNTs and oxMWCNTs_HT

Fourier transform infrared (FT-IR) spectra over the spectral range 400–4000 cm^−1^ were collected with a Perkin–Elmer Spectrum GX infrared spectrometer featuring a deuterated triglycine sulphate (DTGS) detector. Every spectrum was the average of 64 scans taken with 2 cm^−1^ resolution. Samples were prepared as KBr pellets with ca. 2 wt% of sample. Raman spectra were recorded with a Micro-Raman system, RM 1000 RENISHAW, using a laser excitation line at 532 nm (Nd–YAG), in the range of 1000–3500 cm^−1^. A power level of 1 mW was utilized with a 1 μm focusing spot so as to avoid the photodecomposition of the samples. Thermogravimetric measurements were carried out with a Perkin Elmer Pyris Diamond TG/DTA. Samples of approximately 5 mg were heated in air from 25 °C to 900 °C at a rate of 5 °C/min. XPS measurements were applied under an ultrahigh vacuum with a base pressure of 4 × 10^−9^ mbar, using a SPECS GmbH instrument equipped with a monochromatic MgKa source (hv = 1253.6 eV) and a Phoibos-100 hemispherical analyzer. The energy resolution was set to 1.18 eV. Recorded spectra were set with an energy step of 0.05 eV and dwell time of 1 s. All binding energies were referenced with regard to the C1s core level centered at 284.6 eV. Spectral analysis included Shirley or linear background subtraction, and peak deconvolution involving mixed Gaussian–Lorentzian functions was conducted with a least squares curve-fitting program (WinSpec, University of Namur, Belgium). Atomic force microscopy (AFM) images were recorded on silicon wafer substrates using tapping mode with a Bruker Multimode 3D Nanoscope (Ted Pella Inc., Redding, CA, USA).

### 2.5. Cell Culture

Albino Swiss mouse embryo fibroblasts (NIH/3T3, ATCC CRL-1658) and cells derived from Hsp70-transgenic mice overexpressing human heat shock protein 70 (Tg/Tg) were used in this study [30]. The cells were grown in sterile 10-cm-diameter dishes. The medium used for the cultures of both cell lines was Dulbecco’s Modified Eagle’s Medium (DMEM) high-glucose (Sigma-Aldrich D6429, Burlington, MA, USA), supplemented with penicillin/streptomycin (Biowest, L0022-020, Nuaille, Cholet, France), L-glutamine (Biowest, X0550-100, Nuaille, Cholet, France), and fetal bovine serum (FBS) (BIOTECH, P40-37500 PAN). Subculture of the cells was performed up to three times a week and the cells were kept at 37 °C and 5% CO_2_.

### 2.6. Determination of Cellular Viability

For the estimation of cell viability, 96-well plates were used. In each plate, 5000 cells (Tg/Tg or NIH/3T3 cells) were plated with 100 µL of the medium. After 24 h, oxMWCNTs, HT, and oxMWCNTs_HT were added at various concentrations. The plates then were filled with medium to a final volume of 200 µL. Cells were incubated with the substances for 24 or 48 h. At the end of this time, 40 μL of 3-(4,5-dimethylthiazol-2-yl)-2,5-diphenyltetrazolium bromide (MTT tetrazolium salt, stock solution of 3 mg/mL) was added. The cells were incubated again under the same conditions (37 °C and 5% CO_2_) for three hours. The supernatant was then removed, and 100μΙ of DMSO was added. Absorbance at 540 nm and 690 nm was measured with the aid of a spectrophotometer (Multiskan Spectrum, Thermo Fisher Scientific, Waltham, MA, USA). All experiments were performed in triplicate.

### 2.7. Clonogenic Assay

Cells (500 cells/mL) were seeded in 6-well plates, to a final volume of 2 mL per well. After 24 h of incubation, cells were treated with oxMWCNTs (10 and 50 μg/mL), HT (1, 10, 20, and 50 μg/mL), and oxMWCNTs_HT (1 and 20 μg/mL) for an additional 24 h. The supernatant was then removed, and a new medium (2 mL) was added again. The cells were incubated for 8 days with intermediate medium renewal at day 4. On the 8th day, the medium was removed, and cells were rinsed carefully with PBS and stained with a mixture of glutaraldehyde (6.0%) (Thermo Scientific A17876AE, Waltham, MA, USA) and crystal violet (0.5%) (Sigma-Aldrich C3886, Burlington, MA, USA) for 30 min. Plates were then carefully washed and left to dry in normal air at room temperature [31]. Visible colonies were measured using the OpenCFU open-source software [32]. All experiments were performed in triplicate.

### 2.8. Flow Cytometry

#### 2.8.1. Determination of Intracellular ROS Formation

Cells (75 × 10^3^ cells/mL) were plated in 6-well plates and treated with 1 and 20 μg/mL oxMWCNTs, HT, or oxMWCNTs_HT for 24 h. Cells were initially washed with PBS, detached with trypsin, and placed in 2 mL of HBSS (Biosera, LMS2034, Nuaille, Cholet, France). They were then stained with 2′,7′-dichlorofluorescein diacetate (DCF-DA, Sigma-Aldrich D6883, Burlington, MA, USA) solution (2.5 μΜ) for 15 min at 37 °C in the dark. Finally, 1 µg/mL propidium iodide (PI) (Sigma-Aldrich, P4170, Burlington, MA, USA) was added, and the samples were placed on ice and immediately analyzed on a fluorescence-activated cell sorting flow cytometer (Partec ML, Partec GmbH, Germany). For each sample, 10,000 events were measured. All experiments were performed in triplicate.

#### 2.8.2. Cell Cycle

Cells were seeded in six-well plates at a density of 75 × 10^3^ cells/mL before the addition of 20 μg/mL oxMWCNTs, HT, or oxMWCNTs_HT. Twenty-four hours later, the supernatant was discarded and cells were washed twice with PBS, detached with trypsin, and collected with PBS. The cell suspensions were centrifuged for 5 min at 3000 rpm. The supernatants were removed and the cells were resuspended in ice-cold PBS. Centrifugation was re-performed and the pellet was re-suspended with 0.5 mL iced-cold PBS. Following this, 0.5 mL of absolute ethanol was added using the drop-by-drop technique. The samples were then stored at −20 °C for one week. On day 8, the samples were centrifuged and the pellets were resuspended in 1 mL of iced-cold PBS. PI and RNAse-A at similar concentrations (25 μg/mL) of each were added to the samples, followed by incubation at 37 °C for 30 min in the dark. Analysis was performed on a fluorescence-activated flow cytometer (Partec ML, Partec GmbH, Germany). For each sample, 10,000 events were counted. All experiments were performed in triplicate.

### 2.9. Statistical Analysis

Data are expressed as means ± standard deviation. The statistically significant difference between data means was determined by the Student *t*-test. *p* < 0.05 indicated a statistically significant difference (SPSS version 20.0, Statistical Package for the Social Sciences software, SPSS, Chicago, IL, USA). The GraphPad Prism 8 software was used for creating the figures.

## 3. Results

### 3.1. Structural and Morphological Characterization of oxMWCNTs and oxMWCNTs_HT

The samples of the oxidized MWCNTs (oxMWCNTs) and the modified ones with hydroxytyrosol (oxMWCNTs_HT) are shown in Figure 3. For comparison reasons, the spectra of pristine MWCNTs and hydroxytyrosol are presented as well. In the spectrum of the pristine MWCNTs, no intense vibrations were recorded, in contrast to the spectrum of the oxidized ones. More specifically, the bands at 613 cm^−1^ and 1107 cm^−1^ are attributed to the wagging vibrations of the hydroxyl groups and stretching vibrations of the epoxide groups (C-O-C), respectively. Stretching vibrations of C-OH groups are observed at 1413 cm^−1^, while the band at 1618 cm^−1^ is associated with the bending vibrations of water molecules. Additionally, the band at 3436 cm^−1^ corresponds to vibrations of hydroxyl groups [33]. The presence of the aforementioned bands confirms the creation of oxygen functional groups on the surface of the nanotubes and, therefore, the successful oxidation of the material. In the case of oxMWCNTs_HT, the appearance of the two bands in the range of 2800–3000 cm^−1^ is related to the stretching vibration of alkyl groups of hydroxytyrosol molecules. The same bands are detected in the spectrum of HT, which leads to the conclusion that the phenolic compound was successfully attached to the surface of the graphitic walls of the nanotubes. The vibration bands derived from HT attached to the nanotubes were reduced, and the remaining ones shifted slightly compared to the spectra of HT. This is due to the fact that the freedom of some bonds to vibrate is reduced significantly as the functional groups take part in the interactions between the oxMWCNTs and HT.

The Raman spectra of MWCNTs, oxMWCNTs, and oxMWCNTs_HT (Figure 4) depict the intense D, G, and G’_2D_ bands characteristic of pristine and oxidized MWCNTs [34,35]. The Raman spectrum of MWCNTs is similar to those of the paracrystalline carbon black and amorphous biochar, exhibiting an intense and broad D band [36]. In contrast to SWCNTs, *sp*^2^-to-*sp*^3^ hybridization associated with the oxidation or functionalization of MWCNTs does not generally result in drastic changes to their intrinsically intense D bands [36]. This renders the D-to-G band intensity ratio ($\frac{I_{D}}{I_{G}}$) an unreliable parameter for the quantification of the extent of oxidation or functionalization of MWCNTs [37,38]. Instead, the disorder associated with the oxidation or functionalization of pristine MWCNTs is characterized by the intensifying of another weak defect-activated Raman band, the D’ band appearing at ~1600–1620 cm^−1^ as a shoulder of the G band, hence giving the G bands of pristine, oxidized, and functionalized MWCNTs their characteristic weakly asymmetric profiles [39,40].

The D’ bands of MWCNTs, oxMWCNTs, and oxMWCNTs_H T were unveiled upon deconvolution of their corresponding G bands, as can be seen in Figure 4. To achieve this, the Raman spectra were fit to five symmetric Lorentzian peaks corresponding to the D, G, D’, G’_2D_, and D + G bands. The corresponding Lorentzian curve fitting parameters are presented in Table 1. The D’-to-G band intensity ratio ($\frac{I_{D^{\prime }}}{I_{G}}$), which is directly proportional to the defect concentration [41], was used in place of $\frac{I_{D}}{I_{G}}$ to probe the structural transformations associated with the oxidation of MWCNTs and the adsorption of HT onto oxMWCNTs. As can be seen in Table 1, $\frac{I_{D^{\prime }}}{I_{G}}$ increases from 0.16 for MWCNTs to 0.43 for oxMWCNTs. Furthermore, all the bands of oxMWCNTs and oxMWCNTs_HT reveal a significant increase in the full width at half maximum (FWHM) relative to those of MWCNTs. These observations confirm the oxidation of MWCNTs, which is also corroborated by the intensifying of the D + G bands in the Raman spectra of oxMWCNTs and oxMWCNTs_HT. The peculiar decrease in $\frac{I_{D^{\prime }}}{I_{G}}$ to 0.29 for oxMWCNTs-HT can be attributed to the vibrational contribution of the phenolic C=C bonds in HT at ~1610–1620 cm^−1^ [42,43]. For this reason, the mode of adsorption of HT onto oxMWCNTs, whether it be physisorption or chemisorption, can only be identified by the G’_2D_-to-G band intensity ratio ($\frac{I_{G_{2D}^{\prime }}}{I_{G}}$), which is inversely proportional to the defect concentration [41]. As can be seen in Table 1, $\frac{I_{G_{2D}^{\prime }}}{I_{G}}$ decreases from 0.92 for MWCNTs to 0.44 for oxMWCNTs, which further confirms the oxidation of MWCNTs, but it remains at 0.44 in the case of oxMWCNTs_HT, thus confirming that HT is physisorbed.

The thermogravimetric analyses of pristine MWCNTs, oxMWCNTs, and oxMWCNTs_HT are presented in Figure 5. In the case of MWCNTs, one main combustion is observed, starting approximately at 480 °C, which is accompanied by ~100% mass loss of the material. The analysis of oxMWCNTs indicates the existence of three mass losses. The initial weight loss (up to 100 °C) in the order of 9% wt. corresponds to the removal of the naturally adsorbed water molecules, which is indicative of the material’s hydrophilicity. The next mass loss up to 430 °C (~11%) corresponds to the removal of oxygen groups of carbon nanotubes, while the third weight loss is caused by the deformation of the carbon network (~80%). The curve of oxMWCNTs_HT exhibits a weight loss of ~18.5%, which occurs through the decomposition of both the functional groups and the organic compound. The combustion of the graphitic lattice takes place at approximately 480 °C, which is followed by ~69.5% mass loss. Up to 100 °C, a weight mass loss of 12% is observed, which is attributed to the removal of the water molecules. This value is higher than that of the oxidized nanotubes because of the significant hydrophilic nature of HT.

The surface chemical composition, as well as the valence states of oxMWCNTs and oxMWCNTs_HT, was collected by applying X-ray photoelectron spectroscopy (XPS) measurements. The silica 2p and 2s (Si2p and Si2S) photoelectron peaks (centered at 100 eV and 152 eV, respectively) are due to the silica substrate in which the samples were drop-casted and dried from the solution phase. Figure 6a,b display representative XPS surveys’ scan spectra, which indicate the coexistence of carbon and oxygen atoms. The C1s high-resolution photoelectron spectrum is presented in Figure 4 for both oxMWCNTs and oxMWCNTs_HT. The C1s spectrum of oxMWCNTs (Figure 6c) is deconvoluted into five peaks and demonstrates the successful oxidation of the carbon walls. More specifically, at 284.6 eV, we receive the contribution of the C=C and C-C bonds of the graphitic walls, representing 32.7% of the whole carbon peak, while the next fitted peaks are oxygen functionalities created due to oxidation, such as C-O (24.4%), C-O-C (31.5%), C=O (7.9%), and C(O)O (3.5%). After the attachment of hydroxytyrosol, the C1s spectrum differs from that of oxMWCNTs, as displayed in Figure 6d. The more obvious difference derives from the C=C peak at 284.6 eV, where, in the case of oxMWCNTs_HT, the percentage increases significantly to 52.3%. This is due to the fact that the drug, which consists of a higher amount of C-C bonds, is attached to the surface of the tubes. The ratio of C/O is estimated for both cases and is found to be C/O = 2.0 for oxMWCNTs and C/O = 3.1 for oxMWCNTs_HT. The oxygen functionalities remain after the drug interaction, but all of them are shifted from 0.2 to 0.4 eV, and this phenomenon may be due to the weak interaction of these functionalities with the oxygen species of HT. Finally, a new peak observed and centered at 290.3 eV is due to pi–pi* interaction between the aromatic groups of the drug and the graphitic walls of the nanotubes [44]. We can conclude that the type of interaction of the drug and the nanotubes is via pi–pi* interactions of weak van der Walls forces.

Characteristic AFM images of the synthesized carbon nanostructures are depicted in Figure 7. The usual thickness of the oxMWCNTs is 10–12.5 nm, as derived from the topographic height profile (section analysis) (Figure 7a). Moreover, in the AFM images below are shown the modified structures of the oxidized multi-walled carbon nanotubes with HT, oxMWCNTs_HT (Figure 7b). The average thickness of the oxMWCNTs_HT is varied from 19 to 22 nm, which is indicative of the decoration of the oxMWCNTs with the HT. From the topographic height profile, the AFM images (section analysis) can also be used to calculate the thickness of the HT, which ranges between 8 and 12 nm.

### 3.2. Cell Viability

oxMWCNTs had neither dose- nor time-dependent toxic effects against Tg/Tg and NIH/3T3 cells, whose viability remained over 80% after 48 h of incubation (Figure 8). Moreover, oxMWCNTs_HT did not affect cell proliferation in NIH/3T3 cells (Figure 8c,d) but significantly reduced the Tg/Tg cell population, especially at doses higher than 10 μg/mL and exposure for 48 h (Figure 8b). HT exerted a similar dose-dependent effect in both cell lines, with less than 20% surviving after treatment with 100 μg/mL for 24 h (Figure 8).

Comparing the effect of free HT with that of HT bound to oxMWCNTs (oxMWCNTs_HT), we observed a more potent cytotoxic effect for oxMWCNTs_HT in Tg/Tg cells, which was enlarged with time (Figure 9a,b), and a similar effect on cell viability in NIH/3T3 cells (Figure 9c,d).

### 3.3. Ability of Cells to Form Colonies

Treatment of both cells with HT led to the reduced formation of colonies compared to the control (*p* < 0.05). In Tg/Tg cells, the survival fraction was lower than in NIH/3T3 cells, indicating a more potent cytotoxic effect from HT (Figure 10a). Incubation of cells with 50 μg/mL oxMWCNTs had no effect on NIH/3T3 cells’ ability to form colonies but a mild reduction (20%) was recorded for Tg/Tg cells (Figure 10b). The damages provoked by oxMWCNTs_HT also had a higher impact on Tg/Tg cells’ ability to form colonies than on NIH/3T3 cells (Figure 10c). oxMWCNTs_HΤ (20 μg/mL) carrying 2 μg/mL HT allowed 75% of NIH/3T3 cells and 65% of Tg/Tg cells to retain their reproductive integrity (Figure 10c).

### 3.4. Cell Cycle

HT induced an increase in the S phase (control 23.2 ± 0.5% vs. HT 41.0 ± 4.1%) and G2/M phase (control 16.3 ± 7.0% vs. HT 48.6 ± 6.4%) for Tg/Tg cells (Figure 11a) and a major increase in the S phase (control 28.0 ± 2.2% vs. HT 66.1 ± 6.2%) for NIH/3T3 cells at 24 h after exposure (Figure 11b). On the contrary, neither oxMWCNTs nor oxMWCNTs_HT affected cell cycle phases in both cell lines (Figure 11).

### 3.5. Determination of Intracellular ROS Formation

At a concentration of 20 μg/mL, oxMWCNTs did not trigger ROS formation in either of the two cell lines (Figure 12a,b). HT was able to scavenge intracellular ROS at a similar rate in Tg/Tg (−24% at 1 μg/mL and −17% at 20 μg/mL) (Figure 12b) and NIH/3T3 cells (−20% at 1 μg/mL and −16% at 20 μg/mL) (Figure 10a). Interestingly, the effect of oxMWCNTs_HT in ROS reduction was similar to the effect of HT in NIH/3T3 cells (−25% at 1 μg/mL and −22% at 20 μg/mL, *p* < 0.05) (Figure 12a). On the contrary, the oxMWCNTs-HT caused a significant (*p* < 0.05) increase in ROS produced in Tg/Tg cells (24% at 1 μg/mL and 14% at 20 μg/mL) (Figure 12b).

## 4. Discussion

Hydroxytyrosol is characterized by pharmaceutical companies as a molecule of high interest, and ongoing, extensive research on its biological activities and methods to enhance its beneficial effects is being conducted. Thus, the conjugation of HT to a soluble and biocompatible nanomolecule that effectively carries and releases HT to the bloodstream in a controlled manner is of importance.

We demonstrated that the short-term toxicity of oxMWCNTs was extremely low against the two normal cell lines that we used. However, oxMWCNTs functionalized with HT decreased Tg/Tg viability after 48 h of exposure, indicative of time-dependent cytotoxic activity. Long-term toxicity estimated by the clonogenic assay (mimicking physiological conditions) revealed that Tg/Tg cells were more susceptible to oxMWCNTS_HT’s cytotoxic effects than NIH/3T3 cells, which are potentially provoked by the production of ROS. Matching the amount of HT carried by the oxMWCNTs with free HT (0.5–10 μg/mL), similar cytotoxicity was observed in NIH/3T3 cells with a higher reduction in the viability of Tg/Tg cells.

Several nanotechnological approaches have been employed to increase the delivery and efficiency of HT. Recently, Jadid et al. nano-encapsulated HT and curcumin into poly lactide-co-glycolide-co-polyacrylic acid (PLGA-co-PAA) to enhance their anticancer activity against pancreatic cancer cells and observed a significant increase in apoptotic rates [45]. The same nanocarrier was employed successfully by Ahmadi et al. to enhance the anticancer potential of doxorubicin and HT against HT-29 colon cancer cells [46].

CNTs have been used as carriers for several anticancer molecules, anti-inflammatory agents, steroids, etc. [47]. To the best of our knowledge, this is the first time that oxMWCNTs have been functionalized with HT. The precursor molecule of HT, oleuropein, has successfully been absorbed in two positions (inside and outside) of different types of SWCNTs, with the inner position providing a stronger interaction with oleuropein [48]. However, no biological assays were performed. In 2015, a theoretical study was published on the feasibility of using SWCNTs for the delivery of quercetin, a plant flavonoid [49], and in 2022, SW- and MWCNTs were examined in vitro as nanocarriers for the delivery of 7-hydroxy flavone (7-HF), which belongs to the flavone subgroup of flavonoids and is a potent anti-inflammatory agent. In the latter study, the authors measured the cytotoxicity of pristine SW- and MWCNTs and -COOH-functionalized SWCNTs against several normal and cancer cell lines, but not CNTs functionalized with 7-HF [50]. Their results verified our findings that the cytotoxicity of CNTs is differentiated amongst the different cell lines and is potentially related to the uptake mechanism for each cell type [47].

Hsp-70 protein is a molecular chaperone that acts in a large variety of protein folding and remodeling processes in the cell. Hsp-70 overexpression can protect cells from stress-induced apoptosis [51,52]. It is known that phenols and antioxidant molecules can inhibit the expression of chaperones, such as Hsp-70, resulting in the loss of their protective activity in cells [52]. The effect of oxMWCNTs or oxMWCNTs_HT on cells overexpressing Hsp-70 is, for the first time, reported in our study. The reduction of Tg/Tg cell viability by oxMWCNTs_HT could be an indication that oxMWCNTs successfully delivered the antioxidant molecule HT, inside the cells, which in turn negatively affected Hsp-70’s role; however, this hypothesis needs two-step verification: (a) the intracellular delivery of HT by oxMWCNTs and (b) the exact effect of HT on Hsp-70. Further research is warranted to identify the uptake mechanisms and cellular pathways initiated through the use of oxMWCNTs and oxMWCNTs_HT in this specific type of cells.

## 5. Conclusions

In summary, we have efficiently oxidized MWCNTs creating various oxygen functionalities on the external walls of the tubes. The oxMWCNTs interact via physisorption with the HT drug, creating an efficient platform for the delivery of HT. The oxMWCNTs were non-toxic to the cells. However, when acting as a carrier for HT delivery, oxMWCNTS_HT exerted differential effects against the cells, either by producing ROS (Tg/Tg cells) or by scavenging ROS (NIH/3T3 cells). Complementary data regarding the HT release rates from oxMWCNTs, endocytosis mechanisms, and signal transduction pathway activation are required to completely assess the safety and efficiency of these promising nanocarriers.

## Figures and Tables

**Figure 1 nanomaterials-13-00714-f001:**
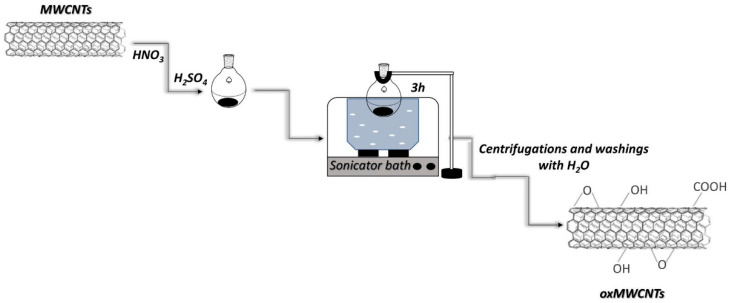
Synthetic procedure of oxMWCNTs.

**Figure 2 nanomaterials-13-00714-f002:**
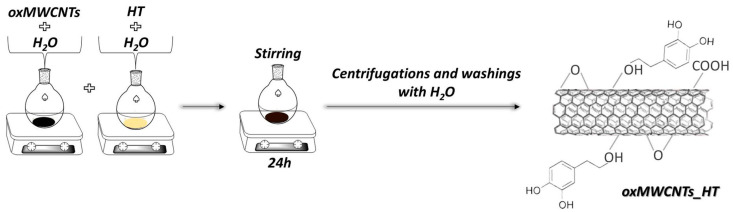
Synthetic procedure of oxMWCNTs_HT.

**Figure 3 nanomaterials-13-00714-f003:**
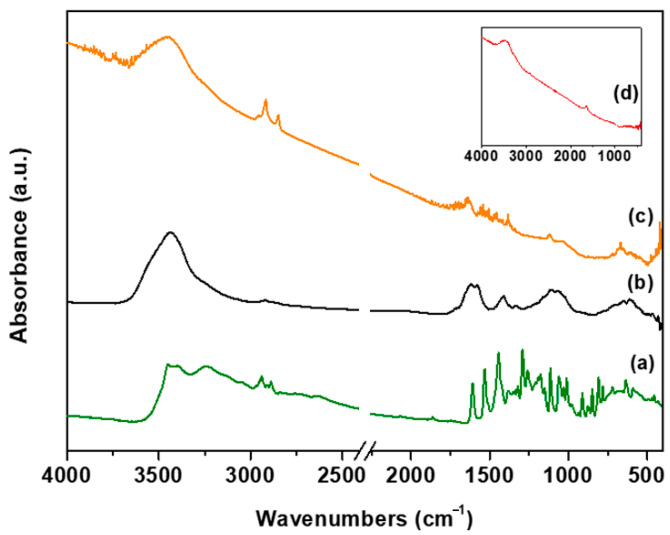
FTIR spectra of (**a**) hydroxytyrosol (HT), (**b**) oxMWCNTs, (**c**) oxMWCNTs_HT, (**d**) pristine MWCNTs.

**Figure 4 nanomaterials-13-00714-f004:**
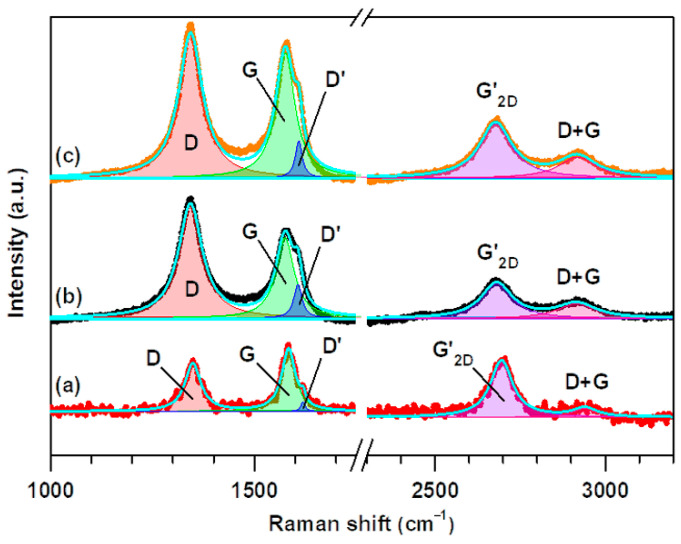
Raman spectra of (**a**) pristine MWCNTs, (**b**) oxMWCNTs, (**c**) oxMWCNTs_HT.

**Figure 5 nanomaterials-13-00714-f005:**
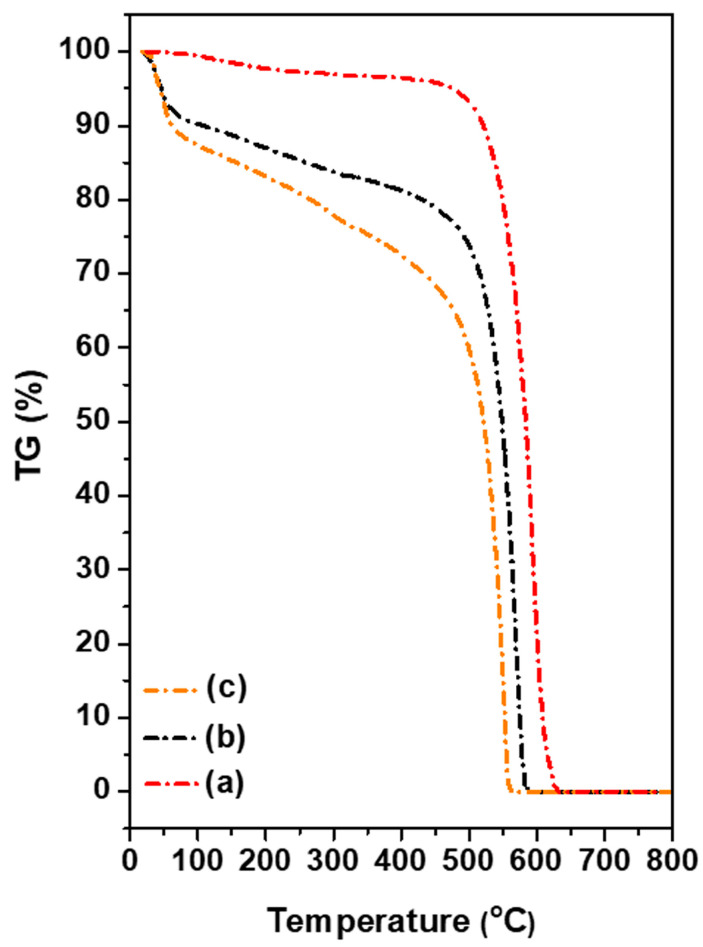
TGA thermographs of (**a**) pristine MWCNTs, (**b**) oxMWCNTs, (**c**) oxMWCNTs_HT.

**Figure 6 nanomaterials-13-00714-f006:**
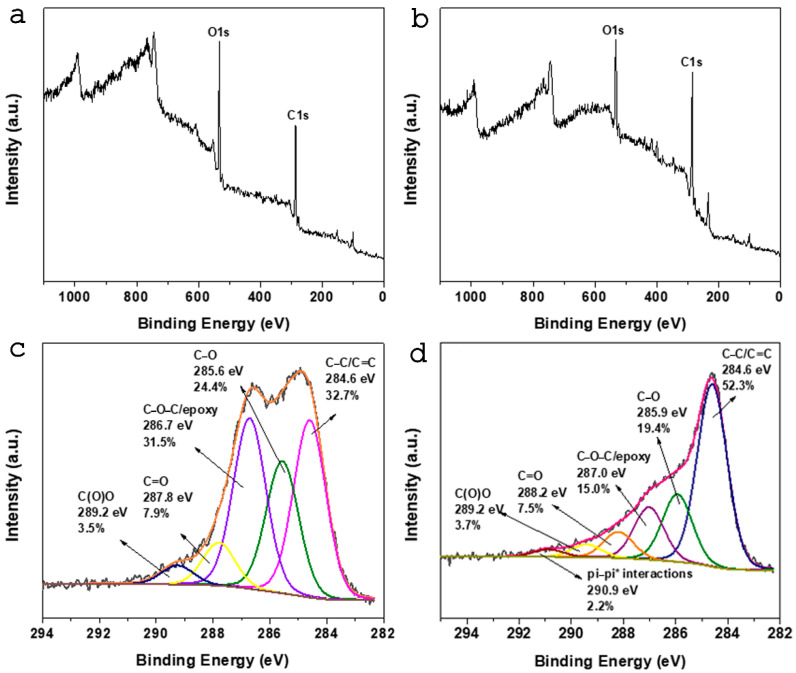
(**a**) XPS survey of oxMWCNTs, (**b**) XPS survey of oxMWCNTs_HT, (**c**) C1s photoelectron spectra of oxMWCNTs, (**d**) C1s photoelectron spectra of oxMWCNTs_HT.

**Figure 7 nanomaterials-13-00714-f007:**
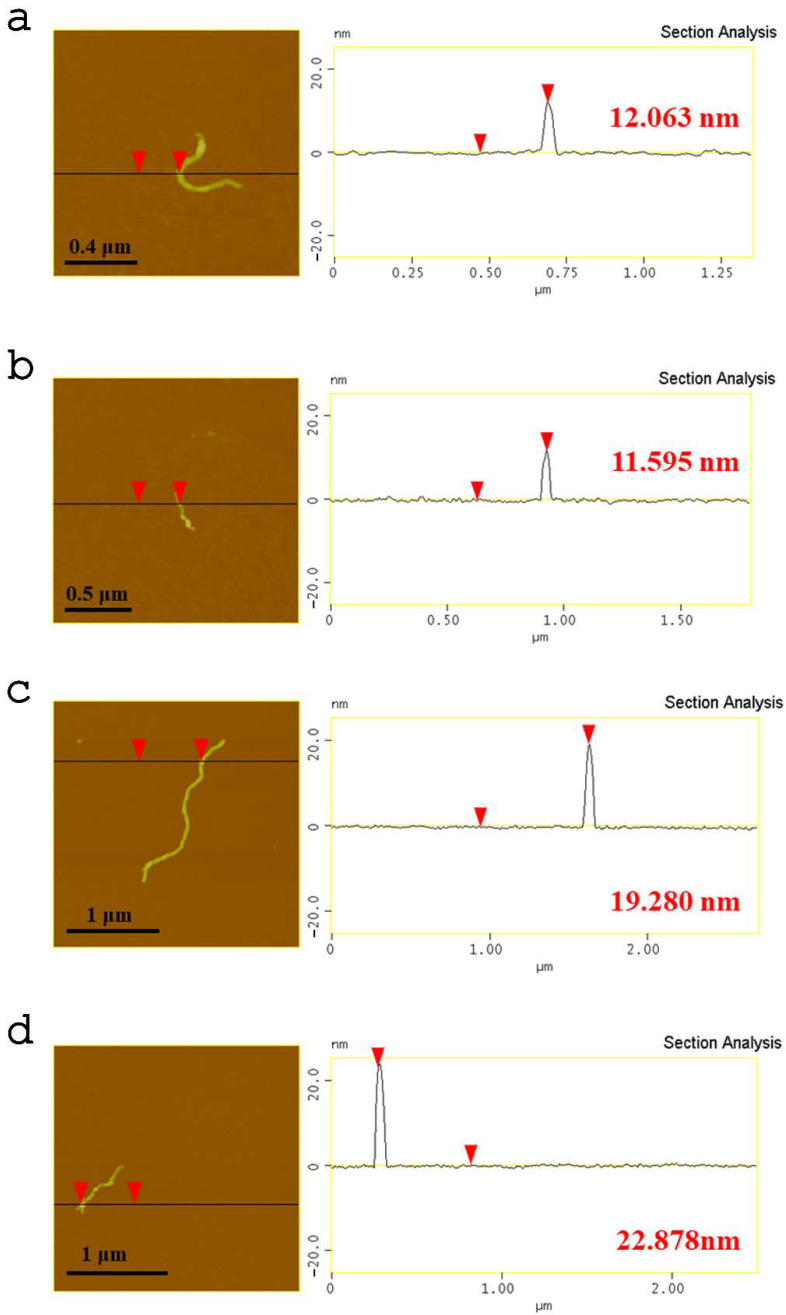
AFM cross-section analysis of (**a**,**b**) oxMWCNTs and (**c**,**d**) oxMWCNTs_HT.

**Figure 8 nanomaterials-13-00714-f008:**
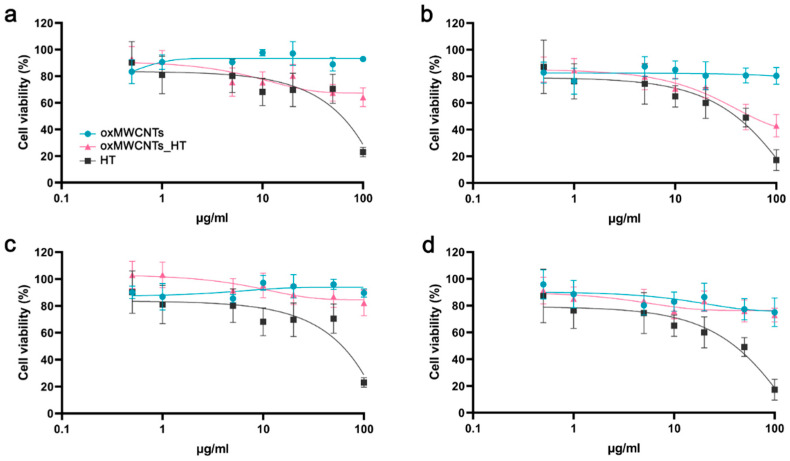
Cell viability of Tg/Tg (**a**,**b**) and NIH/3T3 (**c**,**d**) cells after exposure to oxMWCNTs, HT, or oxMWCNTs_HT for 24 h (**a**,**c**) and 48 h (**b**,**d**).

**Figure 9 nanomaterials-13-00714-f009:**
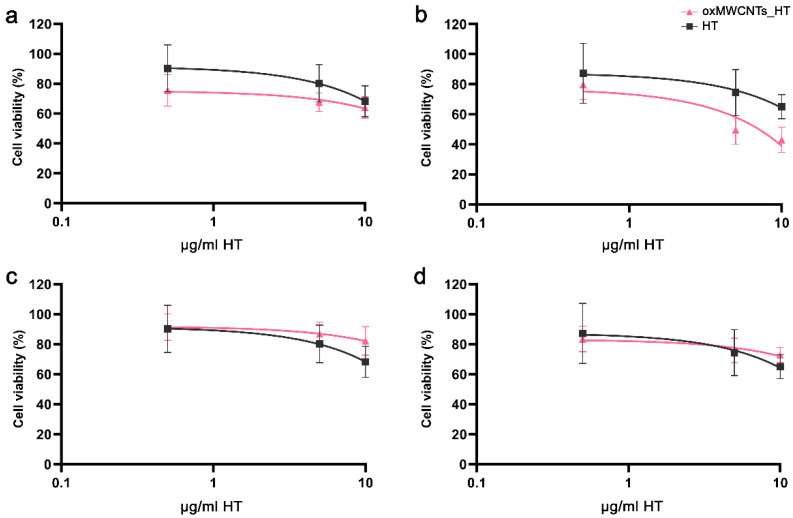
Comparison of oxMWCNTs_HT and HT cytotoxic activity with respect to the net amount of HT. (**a**,**b**) Tg/Tg and (**c**,**d**) NIH/3T3 cell viability after exposure for 24 h (**a**,**c**) and 48 h (**b**,**d**).

**Figure 10 nanomaterials-13-00714-f010:**
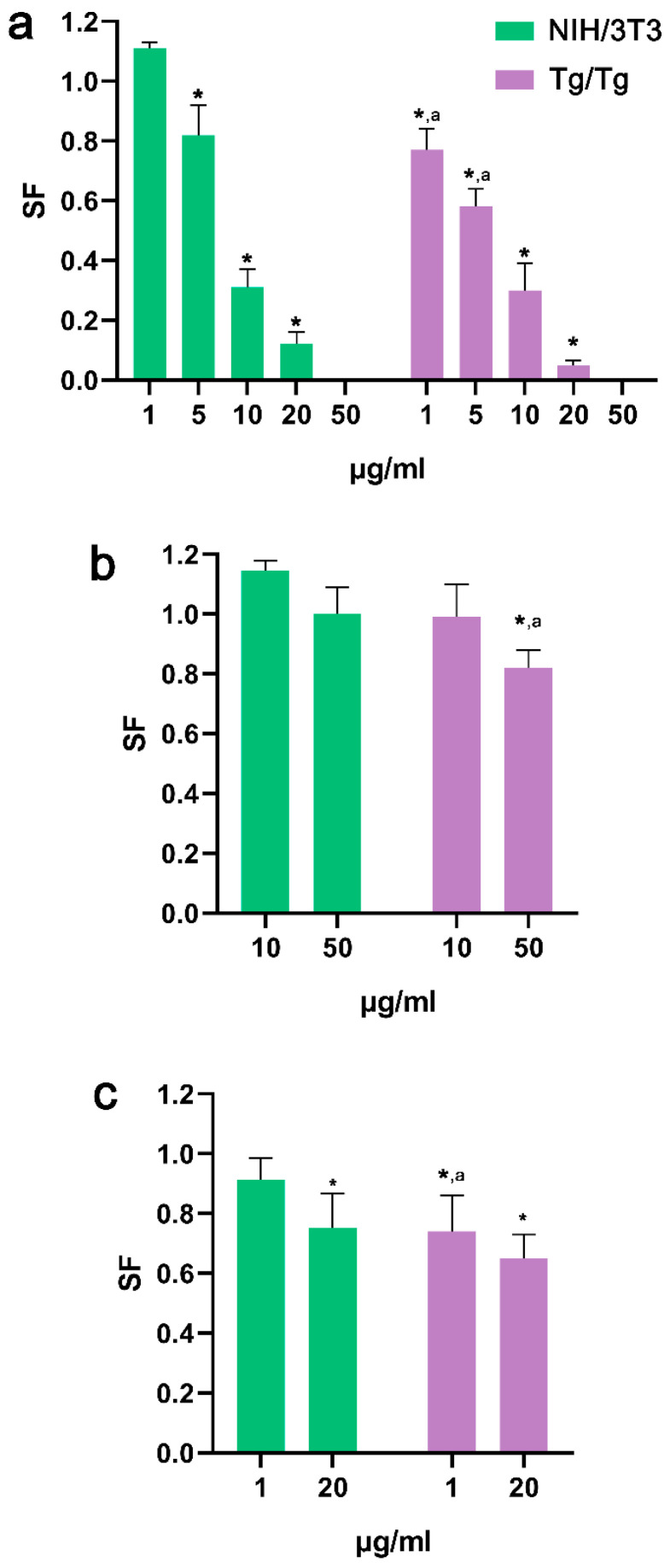
Clonogenic assay of the HT (**a**), oxMWCNTs (**b**), and oxMWCNTS_HT (**c**) in NIH/3T3 and Tg/Tg cells after incubation with various concentrations for 24 h. *, statistically significant difference from control (*p* < 0.05); a, statistically significant difference between the two cell lines (*p* < 0.05).

**Figure 11 nanomaterials-13-00714-f011:**
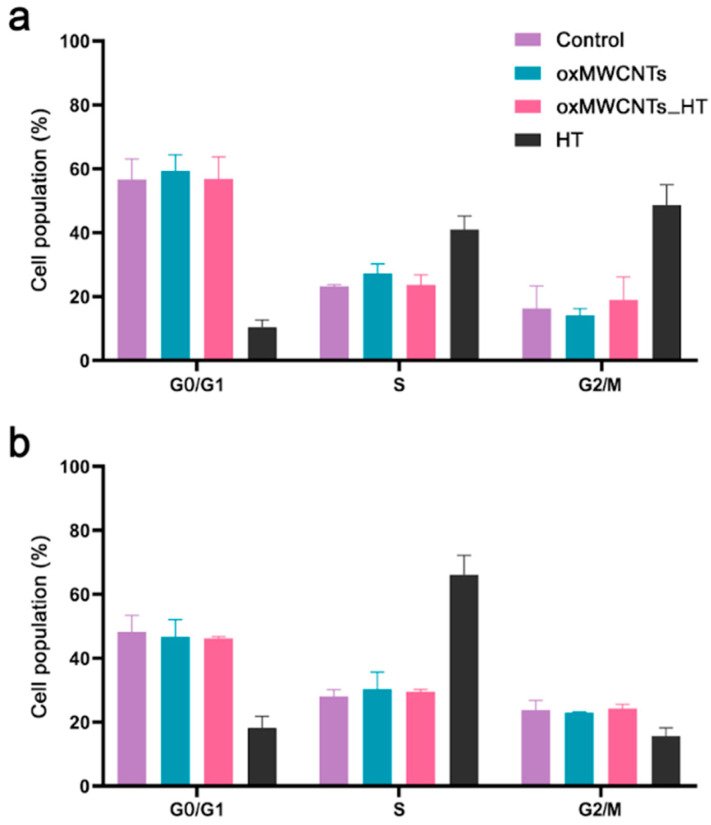
Distribution of phases of the Tg/Tg (**a**) and NIH/3T3 (**b**) cell cycle under the effect of 20 μg/mL oxMWCNTs, oxMWCNTS_HT, and HT for 24 h.

**Figure 12 nanomaterials-13-00714-f012:**
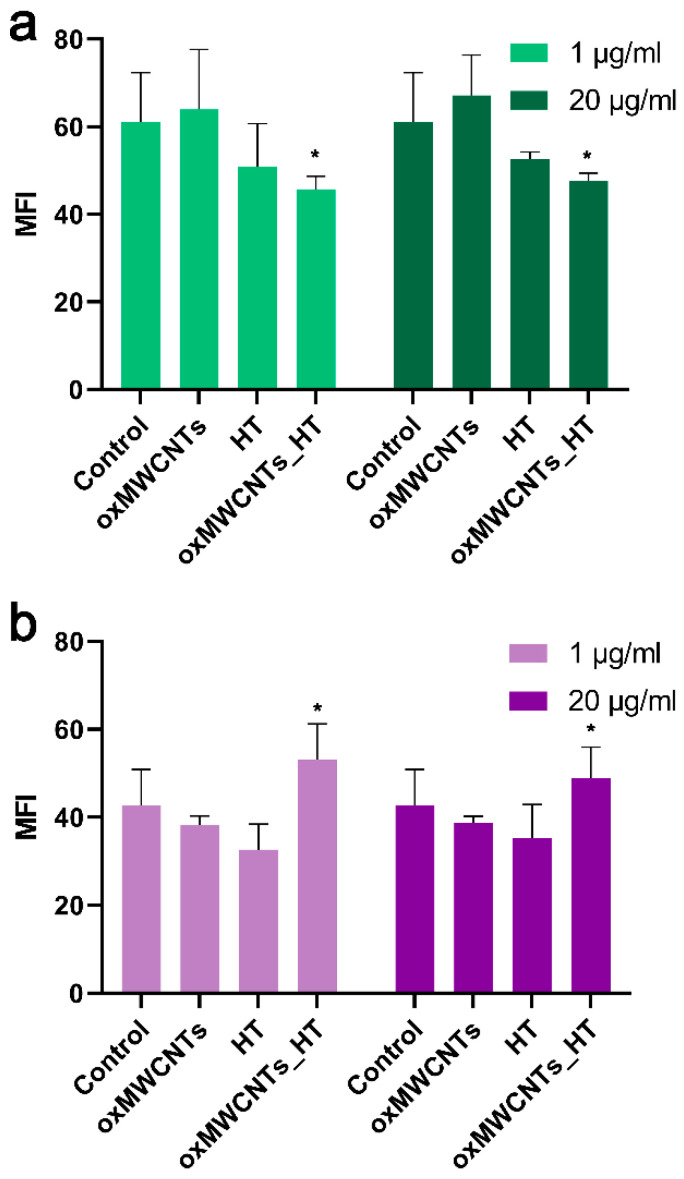
Reactive oxygen species in NIH/3T3 (**a**) and Tg/Tg (**b**) cells after 24 h incubation with 1 and 20 μg/mL of oxMWCNTs, HT, and oxMWCNTs. *, statistically significant difference from control (*p* < 0.05).

**Table 1 nanomaterials-13-00714-t001:** Lorentzian curve fitting parameters for the deconvoluted Raman spectra of (a) MWCNTs, (b) oxMWCNTs, and (c) oxMWCNTs_HT. $\Delta \tilde{\nu }$ = Raman shift/band position; FWHM = full width at half maximum; I = band intensity.

Sample	Bands					$$I_{D^{\prime }}/I_{G}$$	$$I_{G_{2D}^{\prime }}/I_{G}$$
	D	G	D’	G’_2D_	D + G		
**(a)**	$\Delta \tilde{\nu }$ = 1349 cm^−1^ FWHM = 47 cm^−1^ I = 5512	$\Delta \tilde{\nu }$ = 1584 cm^−1^ FWHM = 41 cm^−1^ I = 7108	$\Delta \tilde{\nu }$ = 1619 cm^−1^ FWHM = 12 cm^−1^ I = 1110	$\Delta \tilde{\nu }$ = 2695 cm^−1^ FWHM = 80 cm^−1^ I = 6553	$\Delta \tilde{\nu }$ = 2940 cm^−1^ FWHM = 84 cm^−1^ I = 1168	0.16	0.92
**(b)**	$\Delta \tilde{\nu }$ = 1343 cm^−1^ FWHM = 71 cm^−1^ I = 13,348	$\Delta \tilde{\nu }$ = 1575 cm^−1^ FWHM = 54 cm^−1^ I = 9316	$\Delta \tilde{\nu }$ = 1607 cm^−1^ FWHM = 25 cm^−1^ I = 3974	$\Delta \tilde{\nu }$ = 2681 cm^−1^ FWHM = 111 cm^−1^ I = 4131	$\Delta \tilde{\nu }$ = 2916 cm^−1^ FWHM = 137 cm^−1^ I = 1638	0.43	0.44
**(c)**	$\Delta \tilde{\nu }$ = 1343 cm^−1^ FWHM = 71 cm^−1^ I = 16,823	$\Delta \tilde{\nu }$ = 1576 cm^−1^ FWHM = 57 cm^−1^ I = 14,530	$\Delta \tilde{\nu }$ = 1609 cm^−1^ FWHM = 21 cm^−1^ I = 4259	$\Delta \tilde{\nu }$ = 2677 cm^−1^ FWHM = 115 cm^−1^ I = 6410	$\Delta \tilde{\nu }$ = 2919 cm^−1^ FWHM = 133 cm^−1^ I = 2436	0.29	0.44

## Data Availability

Not applicable.

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
