# Peer review of "Oxidized-Multiwalled Carbon Nanotubes as Non-Toxic Nanocarriers for Hydroxytyrosol Delivery in Cells"

_nanomaterials, 2023, doi:10.3390/nano13040714_

Round 1

Reviewer 1 Report

In this work, multi-walled carbon nanotubes were oxidized effectively and successfully, and used as biocompatible carrier of hydroxytyrosol. The transmission and expression in two different cell lines were investigated. The study provided the possibility for oxidized-multiwalled carbon nanotubes to be a promising nanocarrier in the future, which is interesting. Here are some specific suggestions:

1. In line 174, "75x103 cells/ml" is inconsistent with "75x103cells/ml" above. Please check and correct.

2. In lines 169 and 146, there is no space between units and values, such as 37°Cor 40μI. The same error exists in other parts and should be corrected by checking the manuscript thoroughly.

3. In line 269, whether there is a space between the value of "~ 18.5 %" and the percent sign is inconsistent with other places, please check and unify.

4. The introduction of carbon nanotubes and hydroxytyrosol in the discussion section can be appropriately summarized and transferred to the introduction section at the beginning of the paper.

Author Response

1. In line 174, "75x103 cells/ml" is inconsistent with "75x103cells/ml" above. Please check and correct.

The typo “75x103 cells/ml” was corrected.

2. In lines 169 and 146, there is no space between units and values, such as 37°Cor 40μI. The same error exists in other parts and should be corrected by checking the manuscript thoroughly.

The manuscript was checked thoroughly, and the errors were corrected.

3. In line 269, whether there is a space between the value of "~ 18.5 %" and the percent sign is inconsistent with other places, please check and unify.

Line 269 has been modified accordingly.

4. The introduction of carbon nanotubes and hydroxytyrosol in the discussion section can be appropriately summarized and transferred to the introduction section at the beginning of the paper.

Specific parts from the introduction of the discussion section were transferred to the beginning of the paper and modified accordingly.

Reviewer 2 Report

Journal: Nanomaterials

Article: Oxidized-Multiwalled Carbon Nanotubes as Non-toxic Nanocarriers for Hydroxytyrosol Delivery in Cells

Authors have investigated CNTs as non-toxic carrier for hydroxytyrosol delivery in cells, their material exhibits high-performances and compatibility. The output of the research is interesting and presented results are rigorous and sufficiently meaningful. Besides, the manuscript is well-written. However, various flaws minimize the quality of this research paper and need to be addressed. I recommend Major revisions of this manuscript and advise authors to review below points to enhance the quality of their manuscript before publication in Nanomaterials.

To address,

1) To expend your introduction, I suggest the following references about carbon nanotube engineering, functions and applications (Adv. Mater. 2010, 22, 1672–1688; 10.1002/adma.200901545) (Materials Science and Engineering: B 2021, 268, 115095; 10.1016/j.mseb.2021.115095)

2) While the introduction is well-written, the transition toward the purpose of your study should be much smoother and explained, otherwise the introduction or the objective of your work would have little meaning for the readers.

3) What are the reasons that have pushed you to chose CNTs as a preferential nanocarriers material for hydroxytyrosol delivery?

4) The experimental process could be illustrated.

5) The degree sign could be written as follow “°

6) Figure could be better presented. Especially the figure 5, it is not clear enough for the reader comprehension.

7) Part 2.2, “4” could be written as “four”

8) Correct English mistakes here and there

Author Response

  1. To expend your introduction, I suggest the following references about carbon nanotube engineering, functions and applications (Adv. Mater. 2010, 22, 1672–1688; 10.1002/adma.200901545) (Materials Science and Engineering: B 2021, 268, 115095; 10.1016/j.mseb.2021.115095).

The two proposed references have been added to the manuscript.

  1. While the introduction is well-written, the transition toward the purpose of your study should be much smoother and explained, otherwise the introduction or the objective of your work would have little meaning for the readers.

We have modified our introduction section to emphasize the purpose of our study and make clear the objective for the readers.

  1. What are the reasons that have pushed you to chose CNTs as a preferential nanocarriers material for hydroxytyrosol delivery?

Carbon nanotubes possess a series of excellent properties such as excellent electrical and thermal conductivity, perfect tensile strength combined with high elasticity, chemical stability, and ultrahigh surface area, while its surface can be easily tailored and functionalized. Additionally, the unique shape, alongside the p-p conjugated system and the small nanometer size of the CNTs, generates a series of impressive structural, mechanical, optical, and electronic properties.

Some of these exceptional properties have led MWCNTs as potential effective vehicles for drug delivery.  Attempting to exploit all these advantages but also avoid two main drawbacks of MWCNTs which are a) the negligible solubility in aqueous media and b) the high levels of toxicity of pure MWCNTs, we employed a specific oxidizing method which has shown to present low toxicity levels but also significantly increases the polarity of MWCNTs in water. The high functionality of oxMWCNTs has been demonstrated by the successful functionalization of an anti-inflammatory drug hydroxyryrosol and its study on specific Cells.

  1. The experimental process could be illustrated.

Two figures have been added in the experimental section.

  1. The degree sign could be written as follow “°”

The degree signs have been modified according to the suggestion.

  1. Figure could be better presented. Especially the figure 5, it is not clear enough for the reader comprehension.

The figures in Figure 5 have been modified (vertical arrangement instead of horizontal) to increase its size and make it clear for the reader.

  1. Part 2.2, “4” could be written as “four”.

Number 4 was replaced by “four”.

  1. Correct English mistakes here and there.

Manuscript was checked carefully for any mistakes. All changes are in red font.

Reviewer 3 Report

In the last 5-10 years, successful attempts have been made to use CNT as a carrier of antitumor molecules, anti-inflammatory agents, steroids, and other drugs. Recently, data has been obtained on the use of SWCNT to deliver quercetin and a plant flavonoid, which are potent anti-inflammatory agents. In a peer-reviewed work, several nanotechnological approaches have been used to improve the delivery and efficiency of hydroxytyrosol (HT), including the oxidized form of CNTs - OXMWCNT. OXMWCNTs interact with the HT drug, creating an efficient platform for its delivery, while at the same time not exerting cytotoxic effects. The work was done to a good standard and contains interesting results. It corresponds to the profile of the journal "Nanomaterials" and can be recommended for publication. A few small notes:

1. It is desirable to substantiate the reliability of the conclusions by statistical processing of the results obtained, the evaluation of errors and the number of repeated operations performed.

2. What exactly is meant by AOF, the nature of oxygen intermediates?

3. What is the reason for the gradual increase in absorption in Figs. 1c and 1e, in contrast to Fig. 1a and 1b?

4. There are repetitions in the abstract, discussion and conclusion, which it is desirable to avoid.

Author Response

  1. It is desirable to substantiate the reliability of the conclusions by statistical processing of the results obtained, the evaluation of errors and the number of repeated operations performed.

Our conclusions are based on the statistical analysis of our results. Indeed, we neglected to mention the number of repeated operations performed for each experiment. We added this information (each experiment performed in triplicates) in the methodology section.

  1. What exactly is meant by AOF, the nature of oxygen intermediates?

We apologized but we don’t understand the meaning of the word “AOF”.

  1. What is the reason for the gradual increase in absorption in Figs. 1c and 1e, in contrast to Fig. 1a and 1b?

The absorption of the samples as strongly connected with the thickness of the KBr pellets prepared for the measurement and the concentration of the materials in each pellet.

  1. There are repetitions in the abstract, discussion and conclusion, which it is desirable to avoid.

We checked abstract, discussion and conclusions and we deleted any repeated sentences.
